# Pharmacological Evaluation of Newly Synthesized Cannabidiol Derivates on H9c2 Cells

**DOI:** 10.3390/antiox12091714

**Published:** 2023-09-04

**Authors:** Kitti Szőke, Richárd Kajtár, Alexandra Gyöngyösi, Attila Czompa, Adina Fésüs, Eszter Boglárka Lőrincz, Ferenc Dániel Petróczi, Pál Herczegh, István Bak, Anikó Borbás, Ilona Bereczki, István Lekli

**Affiliations:** 1Department of Pharmacology, Faculty of Pharmacy, University of Debrecen, 4032 Debrecen, Hungary; szoke.kitti@pharm.unideb.hu (K.S.); kajtar.ricsi@gmail.com (R.K.); gyongyosi.alexandra@pharm.unideb.hu (A.G.); czompa.attila@gmail.com (A.C.); fesus.adina@pharm.unideb.hu (A.F.); bak.istvan@pharm.unideb.hu (I.B.); 2Institute of Healthcare Industry, University of Debrecen, 4032 Debrecen, Hungary; 3Department of Pharmaceutical Chemistry, Faculty of Pharmacy, University of Debrecen, 4032 Debrecen, Hungary; lorincz.eszter@pharm.unideb.hu (E.B.L.); petroczi.f.daniel@gmail.com (F.D.P.); herczegh.pal@pharm.unideb.hu (P.H.); borbas.aniko@science.unideb.hu (A.B.); 4Doctoral School of Pharmaceutical Sciences, University of Debrecen, 4032 Debrecen, Hungary; 5National Laboratory of Virology, Szentágothai Research Centre, 7624 Pécs, Hungary; 6ELKH-DE Pharmamodul Research Team, University of Debrecen, 4032 Debrecen, Hungary

**Keywords:** cannabidiol, antioxidant, cannabinoids, simulated hypoxia/reperfusion

## Abstract

Cannabidiol (CBD) is a nonpsychoactive phytocannabinoid that can be found in *Cannabis sativa* and possesses numerous pharmacological effects. Due to these promising effects, CBD can be used in a wide variety of diseases, for instance cardiovascular diseases. However, CBD, like tetrahydrocannabinol (THC), has low bioavailability, poor water solubility, and a variable pharmacokinetic profile, which hinders its therapeutic use. Chemical derivatization of CBD offers us potential ways to overcome these issues. We prepared three new CBD derivatives substituted on the aromatic ring by Mannich-type reactions, which have not been described so far for the modification of cannabinoids, and studied the protective effect they have on cardiomyocytes exposed to oxidative stress and hypoxia/reoxygenation (H/R) compared to the parent compound. An MTT assay was performed to determine the viability of rat cardiomyocytes treated with test compounds. Trypan blue exclusion and lactate dehydrogenase (LDH) release assays were carried out to study the effect of the new compounds in cells exposed to H_2_O_2_ or hypoxia/reoxygenation (H/R). Direct antioxidant activity was evaluated by a total antioxidant capacity (TAC) assay. To study antioxidant protein levels, HO-1, SOD, catalase, and Western blot analysis were carried out. pIC_50_ (the negative log of the IC_50_) values were as follows: CBD1: 4.113, CBD2: 3.995, CBD3: 4.190, and CBD: 4.671. The newly synthesized CBD derivatives prevented cell death induced by H/R, especially CBD2. CBD has the largest direct antioxidant activity. The levels of antioxidant proteins were increased differently after pretreatment with synthetic CBD derivatives and CBD. Taken together, our newly synthesized CBD derivatives are able to decrease cytotoxicity during oxidative stress and H/R. The compounds have similar or better effects than CBD on H9c2 cells.

## 1. Introduction

The annual flowering plant *Cannabis sativa* L. (fam. Cannabaceae) is a source of more than 140 phytocannabinoids that have been in use for medical and recreational purposes for several thousand years [1]. Cannabidiol (CBD) and tetrahydrocannabinol (THC) are the two key bioactive phytocannabinoids found in *C. sativa*. CBD is one of the most abundant and therapeutically relevant components of the medicinal plant, and it is considered completely nonpsychoactive in contrast to THC [2,3]. 

In recent decades, there has been a growing interest in phytocannabinoids identified in *Cannabis sativa*. CBD, a nonpsychotropic cannabinoid, has attracted considerable attention for its multiple bioactivities beneficial to human health. This well-researched bioactive compound has exhibited multiple therapeutic effects, including anti-inflammatory, analgesic, immunomodulatory, antiarthritic, anticonvulsant, neuroprotective, procognitive, antiemetic, anti-anxiety, antipsychotic, and anti-proliferative effects, among others [4,5,6].

The therapeutic potential of CBD has been investigated in numerous clinical trials for the treatment of neurodegenerative, cardiovascular, cancer, and metabolic diseases, whose development is related to redox imbalance, oxidative stress, and inflammation [7]. 

There are several pieces of evidence that prove the therapeutic potential of CBD in a wide variety of cardiovascular diseases, such as hypertension, ischemia/reperfusion (I/R), arrhythmias, and heart failure [8,9,10]. CBD administration reduced platelet aggregation and infarct size, and it was advantageous in the prevention of I/R-induced arrhythmias. The inflammatory response was also reduced after the application of CBD [9,11,12]. CBD-activated adenosine receptors seem to play a protective role during ischemia/reperfusion injury [13]. It may also have a beneficial effect in preventing cardiac damage caused by anticancer antibiotics, such as doxorubicin [14]. 

These multiple favorable effects are due to the pleiotropic mechanisms of CBD. One of these mechanisms is that CBD reduces oxidative conditions and decreases the liberation of reactive oxygen and nitrogen species (ROS/RNS). Among others, the level of superoxide anion is decreased due to elevated superoxide dismutase (SOD) and glutathione peroxidase activity induced by CBD. Other antioxidant enzymes, such as catalase (CAT), which catalyzes the conversion of H_2_O_2_ to water, can protect the cells from oxidative stress [15]. The expression of inflammatory cytokines, such as IL-6 and IFN-γ, and the production of matrix metallopeptidases (MMPs) are also diminished after CBD treatment [7]. Vasorelaxation also plays a major role in preventing severe damage to the cardiac tissue under ischemia, hypertension, or heart failure. CBD elevates nitric oxide synthesis due to stimulation of endothelial nitric oxide synthesis [16]. CBD also has an important role in preserving ionic balance during stressful conditions; for instance, it can modulate the Na^+^/Ca^2+^ exchanger and promote Ca^2+^ storage. CBD administration under H/R also inhibits apoptosis and supports cell survival. In the cell survival, upregulated HO-1 could have a critical role [17]. Numerous reports demonstrate that in contrast to other cannabinoids, CBD has a weak effect on cannabinoid receptors [5]. However, CBD activates transient receptor potential (TRP) channels like TRPA1, TRPV1, and TRPV4, even in nanomolar concentrations [18]. 

Nowadays, there is increasing interest in improving the therapeutic profile, biological activity, and pharmacokinetic properties of CBD, which, however, requires chemical modification of the native structure. Interestingly, while the diverse biological effects of cannabidiol are intensively studied, its chemical transformation is only sporadically addressed [19,20,21,22]. Most of the modifications published so far focus on improving the solubility, absorption, and bioavailability of CBD, and little information is available on how chemical modifications affect the biological activities of the synthetic derivatives compared to the parent compound. The study of derivatized cannabinoids is still in its infancy, and recent literature reviews highlight that there is ample room for the development of new chemically modified cannabinoids and a great need for the biological characterization of these new derivatives [23,24]. 

The known chemical derivatizations of CBD can be classified into two main types: (i) modifications of the monoterpene ring [25] and (ii) modifications of the resorcinol moiety [19,20,21,22]. The latter modifications include oxidative conversion to cannabinoid quinones [19,20] and alkylation [21] of the hydroxyl groups or carbamate formation [21] with various reagents. The resorcinol unit of CBD is sensitive to oxidation and easily transforms into a quinoid structure, which, however, is not stable and can dimerize or undergo further oxidative degradation [19]. At the same time, the resorcinol structure offers the possibility of regioselective introduction of various substituents into the benzene ring, primarily substituents containing an amino group, by Mannich-type reactions [20,22]. Therefore, we decided to follow a new modification route that has not been applied to cannabinoids so far: the functionalization of the aromatic ring of CBD by a Mannich reaction. With Mannich-type reactions, three different substitution patterns can be formed on phenols (Figure 1): the aromatic ring can be functionalized with an alkylaminomethyl group (A), an oxazine (B) [20,22], or an alkoxymethyl group (C). Importantly, the presence of the amino groups in the compound types A and B provides an opportunity for salt formation, which increases water solubility and thus bioavailability. In this work, we prepared 1-1 prototypes of all three substitution patterns achievable by Mannich reactions and investigated their cytoprotective effect and molecular mechanism during oxidative stress and hypoxia and reoxygenation.

## 2. Materials and Methods

### 2.1. General Information

CBD was purchased from https://www.cbdepot.eu (Prague, Czech Republic). Kieselgel 60 F254 (Merck, Darmstadt, Germany) sheets were used for thin-layer chromatography. Detection was performed by immersing into ammonium molybdate-sulfuric acid solution and heating. Silica gel 60 (Merck, Darmstadt, Germany, 0.040–0.063 mm) was used for Flash column chromatography. Bruker Avance II 500 spectrometer (Bruker, Germany) was used for recording the ^1^H NMR (500 MHz), ^13^C NMR (125 MHz), and 2D spectra. Me_4_Si (0.00 ppm for ^1^H) and solvent residual signals were used as reference for chemical shifts. (NMR spectra can be found in the Appendix A.) ESI-TOF MS spectra were recorded by a microTOF-Q type QqTOFMSmass spectrometer (Bruker, Germany). It was used in negative or positive ion mode. MeOH was the solvent, and calibration of the mass spectra was performed using the exact masses of clusters [(NaTFA)_n_ + TFA]^+^ from the solution of sodium trifluoroacetate (NaTFA). DataAnalysis 3.4 software from Bruker was used for the evaluation of the spectra.

### 2.2. Synthesis of CBD1

Formaldehyde (36% in water, 125 μL, 1.5 mmol) and *n*-butylamine (148 μL, 1.5 mmol) were dissolved in methanol (10 mL). The mixture was stirred for 20 min, and then cannabidiol (157 mg, 0.5 mmol) was added. The mixture was stirred for 7 days, the solvent was then evaporated, and the residue was purified by flash column chromatography (hexane/acetone 98:2) to yield CBD1 (188 mg, 89%) as a yellowish syrup.

*R*_f_ = 0.47 (hexane/acetone 95:5); ^1^H NMR (500 MHz, CDCl_3_): *δ* (ppm) 6.25 (s, 1H, aromatic C*H*), 5.90 (s, 1H, O*H*), 5.58 (s, 1H, C-2 C*H*), 4.76 (d, 1H, *J* = 9.6 Hz, H-B C*H*_2_a), 4.66 (dd, 1H, *J* = 9.6 and 1.2 Hz, H-B C*H*_2_b), 4.47 (dd, 1H, *J* = 2.5 and 1.4 Hz, H-9 C*H*_2_a), 4.33 (d, 1H, *J* = 2.3 Hz, H-9 C*H*_2_b), 3.97–3.90 (m, 2H, H-3 C*H* and H-A C*H*_2_a), 3.81–3.74 (m, 1H, H-A C*H*_2_b), 2.71–2.57 (m, 2H, butyl C*H*_2_), 2.43–2.30 (m, 3H, H-1″ C*H*_2_ and H-4 C*H*), 2.27–2.17 (m, 1H, H-6 C*H*_2_a), 2.11–2.03 (m, 1H, H-6 C*H*_2_b), 1.81–1.72 (m, 5H, H-5 C*H*_2_ and H-7 C*H*_3_), 1.66 (s, 3H, H-10 C*H*_3_), 1.56–1.47 (m, 4H, H-2″ and butyl C*H*_2_), 1.40–1.28 (m, 6H, H-3″, H-4″ and butyl C*H*_2_), 0.93 (t, 3H, *J* = 7.3 Hz, H-5″ C*H*_3_), and 0.91–0.86 (m, 3H, butyl C*H*_3_); ^13^C NMR (125 MHz, CDCl_3_): *δ* (ppm) 154.0, 152.5, 147.5, 139.7, 139.6 (5C, quat.), 124.7 (1C, C-2 *C*H), 114.4 (1C, quat.), 111.2 (1C, C-9 *C*H_2_), 109.4 (1C, quat.), 109.2 (1C, aromatic *C*H), 81.6 (1C, C-B *C*H_2_), 51.2 (1C, butyl *C*H_2_), 48.6 (1C, C-A *C*H_2_), 47.0 (1C, C-4 *C*H), 35.3 (1C, C-3, *C*H), 31.90, 39.88, 30.6, 30.4, 29.8 (5C, C-1″, C-2″, C-6 and 2xbutyl *C*H_2_), 28.3 (1C, C-5 *C*H_2_), 23.8 (1C, C-7 *C*H_3_), 22.7, 20.7 (2C, C-4″ and butyl *C*H_2_), 18.8 (1C, C-10 *C*H_3_), and 14.2 (2C, C-5″ and butyl *C*H_3_); and ESI-TOF MS: *m/z* calcd for C_27_H_41_NO_2_Na^+^ was 434.303 [M+Na]^+^ and that found was 434.300.

### 2.3. Synthesis of CBD2

CBD (157 mg, 0.5 mmol) was dissolved in ethanol (10 mL); iminodiacetic acid (1.7 mg, 0.0125 mmol, 0.025 equiv.) and formaldehyde (46 μL, 0.55 mmol, 1.1 equiv.) were added, and the reaction mixture was stirred for a day at 60 °C. Then, the solvent was evaporated, and the residue was dissolved in dichloromethane (200 mL) and washed twice with water (2 × 10 mL). The organic phase was dried over Na_2_SO_4_; it was filtered, followed by the evaporation of the solvent in vacuum. Finally, the residue was purified by flash column chromatography (hexane/EtOAc 99.5:0.5 → 99:1) to yield CBD2 (114.6 mg, 62%) as a yellowish syrup.

*R*_f_ = 0.66 (hexane/acetone 9:1); ^1^H NMR (500 MHz, CDCl_3_): δ (ppm) 8.03 (s, 1H, O*H*), 6.20 (s, 1H, aromatic C*H*), 5.99 (s, 1H, O*H*), 5.59 (s, 1H, H-2 C*H*), 4.73–4.59 (m, 2H, methylene C*H*_2_), 4.51 (s, 1H, H-9 C*H*_2_a), 4.40 (s, 1H, H-9 C*H*_2_b), 4.03 (d, 1H, *J* = 9.6 Hz, H-3 C*H*), 3.56–3.44 (m, 2H, ethyl C*H*_2_), 2.52–2.37 (m, 3H, H-4 C*H* and H-1″ C*H*_2_), 2.28–2.16 (m, 1H, H-6 C*H*_2_a), 2.12–2.03 (m, 1H, H-6 C*H*_2_b), 1.83–1.75 (m, 5H, H-5 C*H*_2_ and H-7 C*H*_3_), 1.70 (s, 3H, H-10 C*H*_3_), 1.74–1.84 (m, 5H, H-5 C*H*_2_ and H-7 C*H*_3_), 1.70 (s, 3H, H-10 C*H*_3_), 1.51–1.41 (m, 2H, H-2″ C*H*_2_), 1.37–1.27 (m, 4H, H-3″ and H-4″ C*H*_2_), 1.24 (q, 3H, *J* = 6.8 Hz, ethyl C*H*_3_), and 0.88 (t, 3H, *J* = 6.8 Hz, H-5″ C*H*_3_); ^13^C NMR (125 MHz, CDCl_3_): δ (ppm) 155.9, 155.7, 147.9, 140.0, 139.5 (5C, quat.), 124.7 (1C, C-2 *C*H), 115.1, 112.0 (2C, quat.), 110.9 (1C, C-9 *C*H_2_), 109.3 (1C, aromatic *C*H), 67.7 (1C, methylene *C*H_2_), 65.3 (1C, ethyl *C*H_2_), 46.7 (1C, C-4 *C*H), 36.0 (1C, C-3, *C*H), 33.3 (1C, C-1″ *C*H_2_), 31.8 (1C, C-3″ *C*H_2_), 31.1 (1C, C-2″ *C*H_2_), 30.5 (1C, C-6, *C*H_2_), 28.2 (1C, C-5 *C*H_2_), 23.8 (1C, C-7, *C*H_3_), 22.7 (1C, C-4″ *C*H_2_), 19.3 (1C, C-10 *C*H_3_), 15.2 (1C, ethyl *C*H_3_), and 14.1 (1C, C-5″ *C*H_3_); and ESI-TOF MS: *m*/*z* calcd for C_24_H_35_O_3_^−^ was 371.258 [M-H]^−^ and that found was 371.255.

### 2.4. Synthesis of CBD3

Formaldehyde (36% in water, 250 μL, 5 mmol) and diethylamine (310 μL, 3 mmol) were dissolved in dioxane (10 mL). The mixture was stirred for 20 min, and then cannabidiol (314 mg, 1 mmol) was added. The mixture was stirred at reflux temperature for 6 h and then at 70 °C overnight. The solvent was evaporated, and the residue was purified by flash column chromatography (hexane/ethyl acetate 9:1) to yield CBD3 (325 mg, 67%) as a yellowish syrup. 

*R*_f_ = 0.56 (hexane/acetone 9:1); ^1^H NMR (500 MHz, CDCl_3_): *δ* (ppm) 5.36 (s, 1H, H-2 C*H*), 4.52 (d, 1H, *J* = 2.9 Hz, H-9 C*H*_2_a), 4.42–4.39 (m, 1H, H-9 C*H*_2_b), 4.06–3.96 (m, 1H, H-3 C*H*), 3.75–3.58 (m, 4H, H-A C*H*_2_), 3.20–3.12 (m, 1H, H-4 C*H*), 2.68–2.45 (m, 8H, ethyl C*H*_2_), 2.44–2.38 (m, 2H, H-1″ C*H*_2_), 2.32–2.22 (m, 1H, H-6 C*H*_2_a), 2.01–1.94 (m, 1H, H-6 C*H*_2_b), 1.73–1.80 (m, 2H, H-5 C*H*_2_), 1.68 (s, 3H, H-7 C*H*_3_), 1.63 (s, 3H, H-10 C*H*_3_), 1.38–1.30 (m, 6H, H-2″, H-3″ and H-4″ C*H*_2_), 1.08 (t, 12H, *J* = 7.2 Hz, ethyl C*H*_3_), and 0.94–0.89 (m, 3H, H-5″ C*H*_3_); ^13^C NMR (125 MHz, CDCl_3_): *δ* (ppm) 150.6, 136.6, 130.8, 116.5, 110.3 (8C, quat.), 127.1 (1C, C-2 *C*H), 109.0 (1C, C-9 *C*H_2_), 52.9 (2C, 2x C-B *C*H_2_), 45.9, 45.8, 45.7 (4C, 4x ethyl *C*H_2_), 44.4 (1C, C-4 *C*H), 36.6 (1C, C-3 *C*H), 32.2, 30.9, 30.6, 30.1, 29.0, (5C, C-1″, C-6, C-5, C-2″, and C-3″ *C*H_2_), 23.6 (1C, C-7 *C*H_3_), 22.6 (1C, C-4″ *C*H_2_), 19.5 (1C, C-10 *C*H_3_), 14.2 (1C, C-5″ *C*H_3_), 11.6, 11.4, 11.4, and 11.3 (4C, 4x ethyl *C*H_3_). ESI-TOF MS: *m/z* calcd for C_31_H_51_N_2_O_2_^−^ was 483.396 [M-H]^−^ and that found was 483.396.

### 2.5. Materials

Cell culture medium, bovine serum, MTT, LDH, and Antioxidant Assay Kit were purchased from Sigma (St. Louis, MO, USA). Stain-free gels and PVDF membrane were obtained from Bio-Rad Laboratories (Hercules, CA, USA). Anti-Heme Oxygenase 1 antibodies (HO-1) were purchased from Abcam (Cambridge, UK). Catalase antibodies and superoxide dismutase 2 (SOD2) were bought from R&D systems (Abingdon, UK).

### 2.6. Cell Culture and Treatment Protocol

H9c2 rat cardiomyocytes were obtained from ATCC, CRL-1446, LGC Standards GmbH Wesel, Germany, and were grown in Dulbecco’s modified Eagle’s medium (DMEM) supplemented with 10% fetal bovine serum and 1% streptomycin-penicillin solution at 37 °C in a humidified incubator consisting of 5% CO_2_ and 95% air. 

During the experiment, 60–70% confluent cells were used. Cells were incubated for 24 h to establish adhesion to the wells. CBD and synthetic derivatives: CBD1, CBD2, CBD3, and CBD were used for cell treatment. They were dissolved in DMSO, then further diluted with medium. 

### 2.7. MTT Assay Cell Viability Assessment

H9c2 viability was measured by MTT assay (3-(4,5-dimethylthiazol-2-yl)-2,5-(diphenyltetrazolium bromide)) on 96-well plates. Briefly, H9c2 cells were plated at density of (3 × 10^3^ cells/well) and grown overnight in a humidified incubator with 5% CO_2_ at 37 °C for 24 h. For the determination of IC_50_ values, cells were treated with the following conditions: fresh culture medium alone (control), fresh culture medium with 0.4% DMSO, and with CBD1, CBD2, CBD3, and CBD at concentrations ranging from 1 μM to 500 μM for 24 h. After that, 20 μL of 5 mg/mL MTT solution was added to each well. The plate was again incubated for 3 h at 37 °C. During incubation, a conversion of the water-soluble yellow dye MTT into an insoluble purple formazan by the action of mitochondrial reductase occurred. Formazan crystals were solubilized by adding 200 μL of isopropanol to each well. The concentrations were determined by optical density at 570 nm. The background absorbance was also recorded at 690 nm, which was subtracted. Cell viability was expressed as a percent of the cell viability of the untreated control cells for the measurement of antioxidant activity against H_2_O_2_-induced oxidative stress cells, which were seeded as described for the IC_50_ measurement. After 24 h, cells were treated with 0.4% DMSO, CBD1, CBD2, CBD3, or CBD. Then, cells also were treated with 125 μM H_2_O_2_ for 24 h. On the next day, cell viability was measured by MTT assay as described previously.

### 2.8. LDH Assay for Cytotoxicity Assessment

A commercial Cytotoxicity Detection Kit (LDH) (Sigma, St. Louis, MO, USA) was used according to the manufacturer’s instructions.

Optimized amounts of H9c2 cells, 5 × 10^3^ cells, were seeded on 96-well plates. After 24 h, the medium was changed to serum-free medium (1% FBS containing medium), and cells were treated with 10 μM of CBD1, CBD2, CBD3, and CBD for 24 h at 37 °C. Then, cells were exposed to 4 h hypoxia and 3 h reoxygenation. To simulate hypoxia, cells were cultured in a hypoxic incubator containing 1% O_2_, 5% CO_2_, and 94% N_2_, and cells were covered with a hypoxic solution (in mM: NaCl 119, KCl 5.4, NaH_2_PO_4_ 1.2, MgCl_2_ 0.5, HEPES 5, MgSO_4_ 1.3, CaCl_2_ 0.9, Na-lactate 20, and 0.1% BSA, pH 6.4). During reoxygenation, cells were covered with a normoxic solution (in mM: NaCl 125, KCl 5.4, NaH_2_PO_4_ 1.2, MgCl_2_ 0.5, HEPES 20, MgSO_4_ 1.3, CaCl_2_ 1, glucose 15, taurine 5, creatine-monohydrate 2.5, and BSA 0.1%, pH 7.4) and kept in a standard CO_2_ incubator (5% CO_2_ and 95% air). At the end of the reoxygenation, 100 μL/well supernatant was removed carefully and transferred to a new 96-well plate, and 100 μL reaction mixture (freshly prepared) was added to each well and incubated for up to 30 min at room temperature. Then, the absorption of the released LDH was measured with a microplate reader at 490 nm, and the background was measured at 690 nm. To determine LDH activity, the 690 nm absorbance value was subtracted from the 490 nm absorbance value. All experiments were performed in triplicates. During the calculation, low control was the normoxia control, and high control was Triton-X treated group. 

### 2.9. Trypan Blue Staining for Cell Viability

In order to investigate cell viability, 7 × 10^4^ cells were cultivated into 6-well plates. After 24 h, cells were switched from serum-supplemented medium to serum-free medium. Cells were treated with 10 μM of CBD1, CBD2 and CBD. After 24 h treatment, cells were subjected to 4 h simulated hypoxia and 3 h simulated reoxygenation. During hypoxia, cells were covered with hypoxic solutions, which were collected at the end of hypoxia and changed to serum-free medium. At the end of reoxygenation, trypan blue staining method was used. Cells were collected and dissolved in 50 μL of PBS. Next, 10 μL of cell suspension was mixed with 10 μL of trypan blue solution and incubated for 5 min at room temperature. A total of 10 μL of cell–trypan blue solution mixture was applied to glass hemocytometer, and viable and nonviable cells were counted. Viable cells have a clear cytoplasm, whereas nonviable cells have a blue cytoplasm. The live cell count was divided by the total cell count and was multiplied by 100, the percentage viability. All experiments were performed in duplicates.

### 2.10. Antioxidant Assay

Free radical scavenging activity was assessed by Total Antioxidant Capacity (TAC) assay. The method was based on the scavenging of the 2,2′-azinobis-(3-ethylbenzothiazoline-6-sulfonic acid) (ABTS) radical (ABTS^•^) and converting it into a colorless product. The degree of decolorization induced by a compound is related to that induced by Trolox, giving the TEAC value. Antioxidant activity was expressed as Trolox equivalent antioxidant capacity (TEAC). Briefly, 10 μL of Test Sample, 20 μL of Myoglobin Working Solution, and 150 μL of ABTS Substrate Working Solution were added to each well. They were incubated for 5 min at room temperature. Next, 100 mL of Stop Solution was added. Finally, the endpoint absorbance at 405 nm was read using a plate reader. The water-soluble derivative of vitamin E—Trolox (6-hydroxy-2,5,7,8-tetramethylchroman-2-carboxylic acid)—was used as a reference, and the results were expressed as TEAC values. All experiments were performed in duplicates and repeated two times.

### 2.11. Protein Isolation and Western Blot Analysis

After a pretreatment with a 10 μM concentration of CBD1, CBD2, and CBD, followed by 4 h hypoxia/3 h reoxygenation period, proteins were isolated from H9c2 cells for Western blot. Cells were collected; then, the proteins were isolated, and concentration was determined based on the methods reported previously by Gyongyosi et al. [26,27]. Laemmli buffer was added and boiled for 10 min. Western blot analysis was carried out according to the ChemiDoc Touch Imaging System protocol (Bio-Rad Laboratories, Hercules, CA, USA), and the protocol was followed step by step as we previously published [27]. The primary antibodies used in this study included catalase (CAT), hemoxigenase-1 (HO-1), and superoxide dismutase (SOD). The expression of protein of interest was normalized against the total amount of protein in each lane. The results were evaluated by Image Lab 5.2.1 software (Bio-Rad Laboratories, Hercules, CA, USA). Intensity of all proteins in the lane was measured, and Total Protein Normalization was applied during the evaluation. This type of normalization eliminates the need to use housekeeping protein [28]. 

### 2.12. Data Analysis

All data were analyzed by GraphPad Prism version 7 (GraphPad Prism Software, La Jolla, CA, USA). The unpaired *t*-test was used for comparisons between two groups. Values presented in this study were expressed as mean ± SEM. Error bars on the figures represent standard error of the mean (S.E.M.). The level of statistical significance was * *p* < 0.05, ** *p* < 0.01, *** *p* < 0.001, and **** *p* < 0.0001.

## 3. Results

### 3.1. Design and Synthesis of CBD Derivatives

According to the literature, compounds having resorcinol-type structures can be modified by Mannich-type reactions in the presence of an amine and an aldehyde, generally formaldehyde [20]. It is noteworthy that such a reaction has not yet been performed on cannabinoids. Generally, in the Mannich reaction, the aromatic ring is functionalized with an alkylaminomethyl group. However, if the amine component is iminodiacetic acid, an alkoxymethyl side chain can be introduced into the position next to the phenolic OH group using alcohol as a reagent [29]. With these types of Mannich reactions, different types of products can be achieved from CBD; therefore, we planned and performed modifications shown in Figure 2. In the reaction of CBD with n-butylamine and formaldehyde, an oxazine derivative (CBD1) bearing a butyl side chain was produced, while the reaction with diethylamine resulted in a disubstituted diethylamino methyl derivative (CBD3). Using iminodiacetic acid as a catalyst in ethanol, an ethoxymethyl side chain was introduced next to the phenolic OH group, resulting in CBD2 as the product (Figure 2 and Appendix A).

### 3.2. Safety Evaluation of Synthetic Cannabidiol Derivates, IC_50_ Determination

Firstly, the cytotoxicity of the new derivatives was determined by the MTT assay (Figure 3). H9c2 cells were treated with CBD1, CBD2, CBD3, and CBD in increasing concentrations: 1 µM, 3 µM, 10 µM, 100 µM, 300 µM, and 500 µM, respectively. As depicted in Figure 2, the cell viability of the control group was 100%, and the new compound and CBD had no toxic effect in concentrations from 1, 3, 10, to 30 µM. Based on the outcome of the MTT assay, the half-maximal inhibitory concentration (IC_50_) was determined using GraphPad. As depicted in Figure 2, pIC_50_ values are CBD1: 4.113, CBD2: 3.995, CBD3: 4.190, and CBD: 4.671.

### 3.3. Antioxidant Activity against H_2_O_2_-Induced Oxidative Stress

To study the activity of the molecules against oxidative stress, H9c2 cells were challenged by H_2_O_2_. Figure 4 demonstrates that the 24 h treatment with 125 μM of H_2_O_2_ produced significant cell death, as cell viability decreased to 74%, whereas the cell viability for the control group was 100%. However, pretreatment with CBD1, CBD2, CBD3, and CBD in 1, 3, and 10 μM concentrations slightly enhanced the viability of H_2_O_2_-challenged cells. The highest toxicity level decrement was observed with the 3 μM concentration of CBD1, CBD2, and CBD3. 

### 3.4. The Effect of Synthetic CBD Derivates and CBD on Cell Viability during Stimulated Hypoxia/Reoxygenation

Since oxidative stress plays a major role in ischemia/reperfusion injury, we tested the effect of the new compounds under hypoxia/reoxygenation. In this study, H9c2 cells were treated with 10 μM synthetic CBD derivatives and CBD and then exposed to 4 h simulated hypoxia followed by 3 h simulated reoxygenation. In the normoxic group, the control group cell viability was 86%, and in the treated groups, it was between 87 and 91% (Figure 5A). After H/R in the control group, the mortality was significant, at around 60%. The H9c2 cell viability decrement was about 23% compared to normoxic controls. However, pretreatment with 10 μM of CBD1, CBD2, and CBD can protect against H/R-induced cell deaths. Cells exposed to CBD2 showed significantly greater cell viability compared to the control group. In the case of CBD2 pretreatment, in the normoxic group, cell viability was 91%, while in the hypoxic group, cell viability was 89%.

To further confirm our trypan blue exclusion assay’s results, LDH release was determined. LDH level of the normoxic control was calculated to be 0%. CBD1, CBD2, and CBD in 3 and 10 μM concentrations also have no cytotoxicity. Under H/R, the control group cytotoxicity % increased to 30%, indicating the toxic effect of H/R. Pretreatment with CBD1, CBD2, and CBD in 3 and 10 μM concentrations significantly suppressed LDH liberation and cell death. As depicted in Figure 5B, we measured a significantly lower cytotoxicity in cells treated with CBD1 (14%) and CBD2 (7%) and undergoing H/R in comparison with the control group (30%). 

### 3.5. Antioxidant Capacity

The direct free radical scavenging capacity of CBD was studied using the ABTS radical assay. As Figure 6 shows, 10 μM of CBD possessed the greatest direct antioxidant capacity, followed by CBD2. CBD1 also has some antioxidant activity. However, according to the TEAC, CBD3 showed very poor antioxidant activity.

### 3.6. Antioxidant Protein Levels

Furthermore, we studied the expression of some endogen antioxidant proteins. Pretreatment with CBD1, CBD2, and CBD has significantly increased the expression of catalase under normoxic and hypoxic circumstances. Another antioxidant enzyme, SOD expression level, also showed a slight elevation due to the pretreatment. As shown in Figure 7, cells exposed to CBD1 and CBD2 have significantly increased expression levels of HO-1; however, CBD pretreatment showed a slight alteration during normoxic conditions. Under hypoxic circumstances, CBD1 and CBD2 pretreatment induced HO-1 expression alteration. 

## 4. Discussion

Increased oxidative stress plays a fundamental role in different cardiovascular diseases, including ischemia/reperfusion injury. In this study, we studied the effects of newly synthesized CBD derivatives, CBD1, CBD2, and CBD3, compared with the effect of CBD on cell viability under oxidative stress and simulated H/R. Furthermore, the direct and indirect antioxidant properties of the molecules were also studied. During the development of any new molecule, it is essential to check the cytotoxicity. The safety profile of synthetic CBD derivatives was evaluated by MTT assay. Our results indicated that the new derivatives are safe, and the cytotoxicity is similar to the original molecule, CBD. Previously, M. Rajesh and co-workers demonstrated that CBD treatment was able to prevent oxidative–nitrative stress in diabetic myocardium [10]. Here, we observed that the treatment with low-dose CBD derivatives showed a slight decrement in H_2_O_2_-induced redox imbalance; however, the difference was not statistically significant. S. K. Walsh et al. found that CBD administration protects against myocardial I/R injury, evidenced by the decreased infarct size in the pre-ischemia CBD-treated group and in the pre-reperfusion CBD-treated groups, respectively [9]. In line with the literature, we found that CBD treatment significantly attenuated H/R-induced LDH liberation. Moreover, our new CBD derivatives, especially CBD2, exhibited an even better protective effect than native CBD and increased cell viability, evidenced by trypan blue staining.

The free radical scavenging ability of CBD is thought to contribute to its biological effect. In recent years, several studies have aimed to investigate the free radical scavenging capacity of CDB and other phytocannabinoids originating from *Cannabis sativa* [30,31,32]. However, recently Boulebd and colleagues showed that the antioxidant activity of CBD depends on the media of the experiments. They suggested that in the aqueous phase at physiological pH, CBD exhibits significant antioxidant activity, but in physiological lipid media, the effect is almost diminished [33]. In our study, we used the ABTS assay to compare the antioxidant properties of CBD and its derivatives based on the following reasons. ABTS is a commonly used method to measure the antioxidant activity of CBD and phytocannabinoids, and based on the previous results [30,31,32] and our own observation, it is more sensitive to weaker antioxidants. In previous studies using different assays, the authors always found antioxidant activity for CBD. Although there were differences between the antioxidant intensity based on the chosen assays, in all cases, the antioxidant activity of CBD was comparable with vitamin E [34]. We chose the ABTS assay and compared our new molecules to the parent molecule. Previously, it has been suggested that the antioxidant activity of CBD and phytocannabinoids relies on the phenolic groups, which can easily be converted to quinoid form [34,35]. Our results are in line with the literature; we found no antioxidant activity of CBD3, since the para substituents of phenolic groups prevent the transformation of the molecule into the quinoid form. Also, CBD1 exhibits moderate antioxidant activity, which may relate to the [1,3]-oxazine group. However, the antioxidant effect of the new derivatives is lower than that of the parent molecule, CBD.

The role of oxidative stress has been implicated as an underlying mechanism of different diseases, including I/R and the generation of ventricular arrhythmia, which is related to imbalanced ion homeostasis. Thus, we further studied components of the endogen antioxidant system. Increased levels of SOD and CAT can protect against ROS-induced damage [36]. CBD treatment enhanced the expression levels of the main endogenous antioxidant systems, such as catalase and superoxide dismutase [37]. Interestingly, Usami and colleagues, in an ex vivo experiment using mouse hepatic microsomes, found that CBD and its hydroxy-quinone derivatives produced by CYP3A11 reduced the activity of SOD and catalase [38]. On the contrary, in another ex vivo model in which FeSO_4_ was employed to induce oxidative stress in isolated hepatic tissue, leading to decreased activity of SOD and catalase, CBD treatment enhanced the antioxidant defense, including SOD and catalase activity [39]. The authors suggested that this contributes to the hepatoprotective effect of CDB and *Cannabis sativa*. Our results showed that mainly, the catalase level was increased, even in the normoxic group, which could contribute to the beneficial effects of CBD1 and 2. Similarly, enhanced catalase activity was found in mice hippocampal tissue after CBD treatment in a pilocarpine-induced seizure rat model [40].

The protective role of HO-1 against I/R injury has been extensively studied. Earlier, we have shown that overexpression of HO-1 by different natural products, including sour cherry seed extract or garlic, contributes to the protective properties of the natural products [41,42]. Furthermore, Böckmann and Hinz found that CBD increases the level of HO-1 in HUVEC cells in a dose-dependent manner [17]. The authors demonstrated that CBD activates the Nrf2/HO-1 pathway, leading to the activation of autophagy and suppression of apoptosis. However, in higher concentrations, this protective effect is lessened, and apoptosis and cell death may occur. Recently, Zhang and colleagues demonstrated that activation of HO-1 contributes to the cardioprotective effect of CBD in myocardial injury caused by exhaustive training [43]. Based on molecular docking results, the role of the Keap1/Nrf2/HO-1 axis was suggested. In agreement with the literature, in this study, we have shown that the new derivatives also enhance the level of HO-1, which contributes to their protective effects. There are some limitations to the current study. There were no direct ROS measurements during the H/R experiments. However, it is not likely that 4 h of hypoxia and 3 h of reoxygenations are not enough to cause sufficient injury to the cells, since it is a well-established model. Furthermore, additional studies with the current derivatives and new similar molecules need to be performed to accurately demonstrate precise structure–activity relationships. 

In conclusion, we successfully used Mannich-type reactions to substitute CBD on an aromatic ring, resulting in new non-toxic derivatives. CBD and our synthetic molecules (CBD1, CBD2) are capable of decreasing oxidative stress and protecting against H/R, therefore possibly preventing I/R injury. Under our experimental circumstances, CBD2 was the most promising molecule. However, further in vivo studies are required to confirm the results of the current study.

## Figures and Tables

**Figure 1 antioxidants-12-01714-f001:**
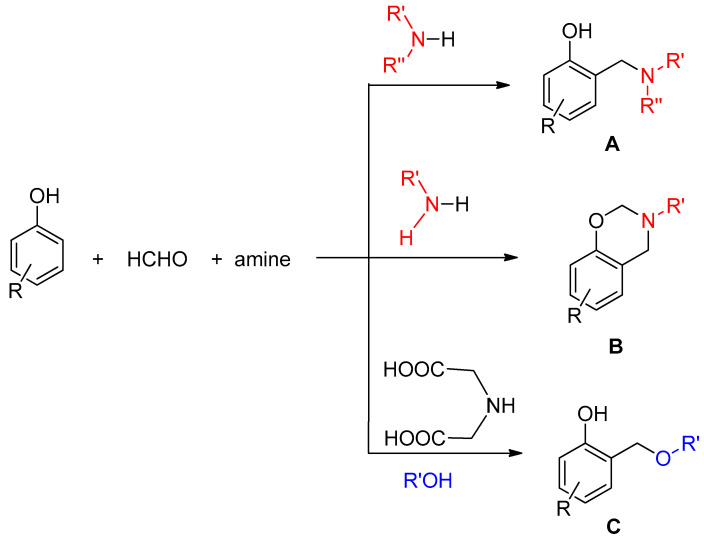
Different functionalizations of phenols by Mannich-type reactions resulting A, B or C types of products.

**Figure 2 antioxidants-12-01714-f002:**
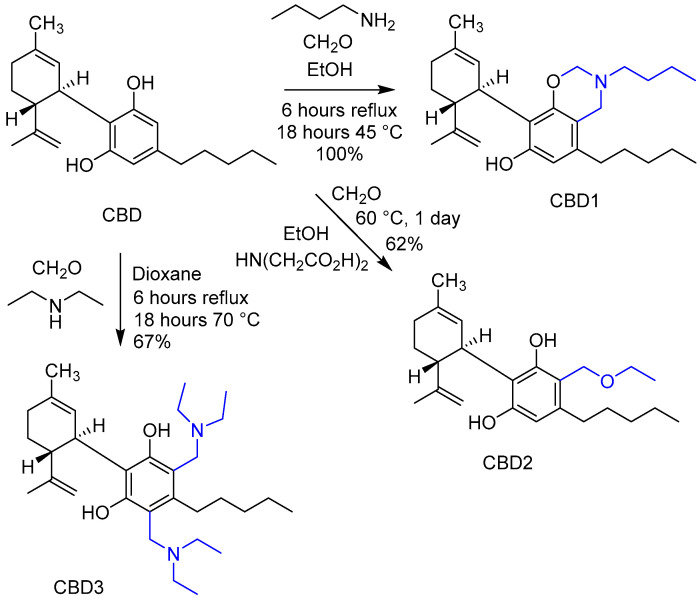
Synthetic modifications of CBD.

**Figure 3 antioxidants-12-01714-f003:**
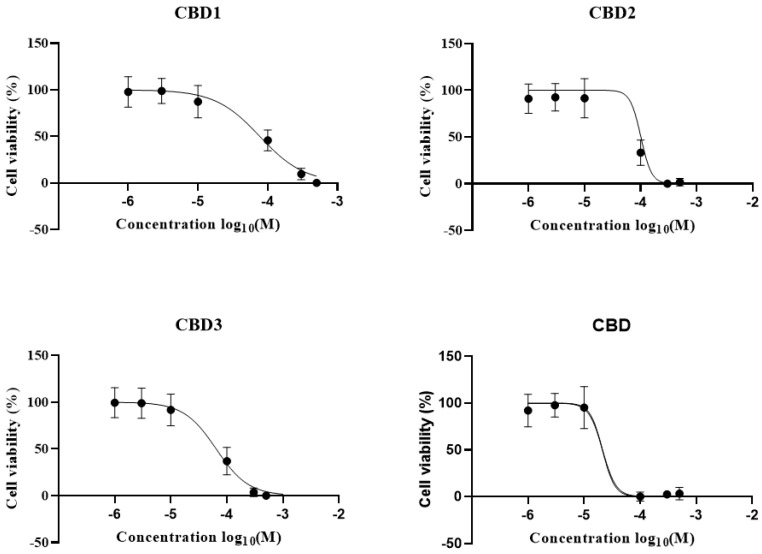
Cell viability assay using the MTT method. H9c2 cells were treated with different concentrations of synthetic CBD derivatives and 1 µM, 3 µM, 10 µM, 100 µM, 300 µM, and 500 µM of CBD for 24 h, and 0.4% DMSO was used as vehicle-treated control. The measurements were carried out in quintuplicate. Columns represent the mean ± SEM (n = 3–6).

**Figure 4 antioxidants-12-01714-f004:**
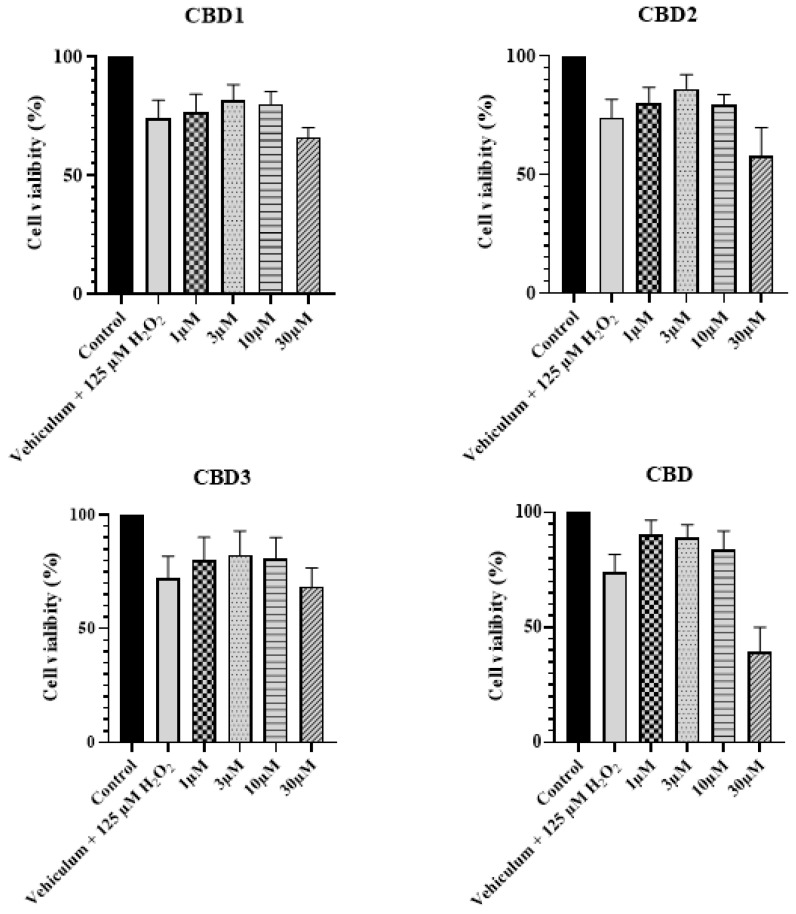
Effect of synthetic CBD derivatives and CBD on cell viability after 125 μM H_2_O_2_ treatment on cell viability evaluated by MTT assay. Data shown are mean ± SEM (n = 6).

**Figure 5 antioxidants-12-01714-f005:**
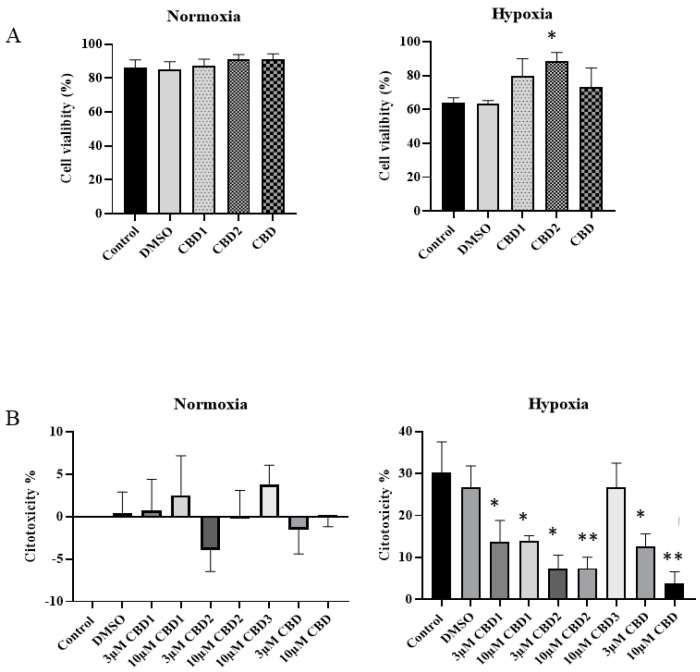
Cell viability evaluated after H/R on cells pretreated with the synthetic CBD derivatives and CBD. (**A**) Cytotoxicity assessed by trypan blue exclusion test. Trypan blue dye exclusion assay was performed to study the protective effect of the treatments during hypoxia. The numbers of viable and nonviable cells were counted manually using hemocytometer. Data are represented as mean ± SEM. The measurements were carried out in duplicate (n = 3 experiments). (**B**) Cytotoxicity tested by LDH assays. Data are represented as mean ± SEM. The measurements were carried out in triplicate. n = 4–6 experiments. * *p* < 0.05 and ** *p* < 0.01.

**Figure 6 antioxidants-12-01714-f006:**
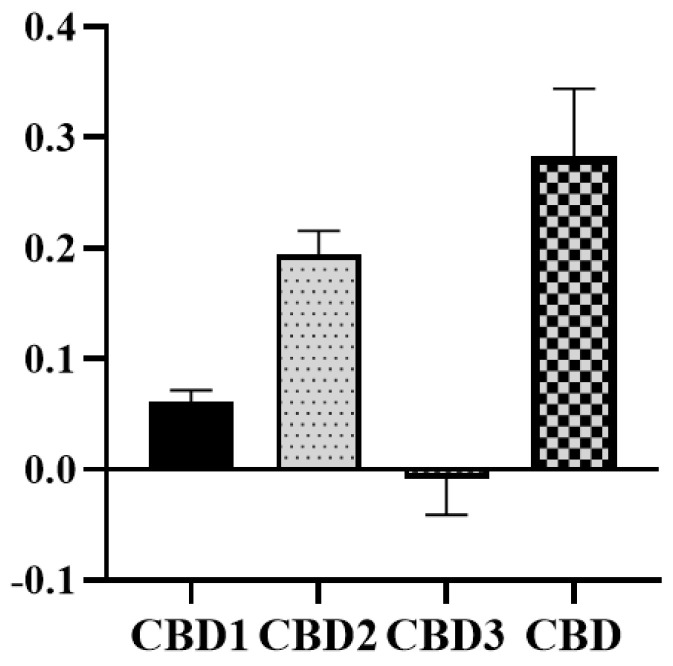
Radical scavenging capabilities of synthetic CBD and CBD by TEAC. The measurements were carried out in duplicate columns. Data represent the mean ± SEM (n = 2).

**Figure 7 antioxidants-12-01714-f007:**
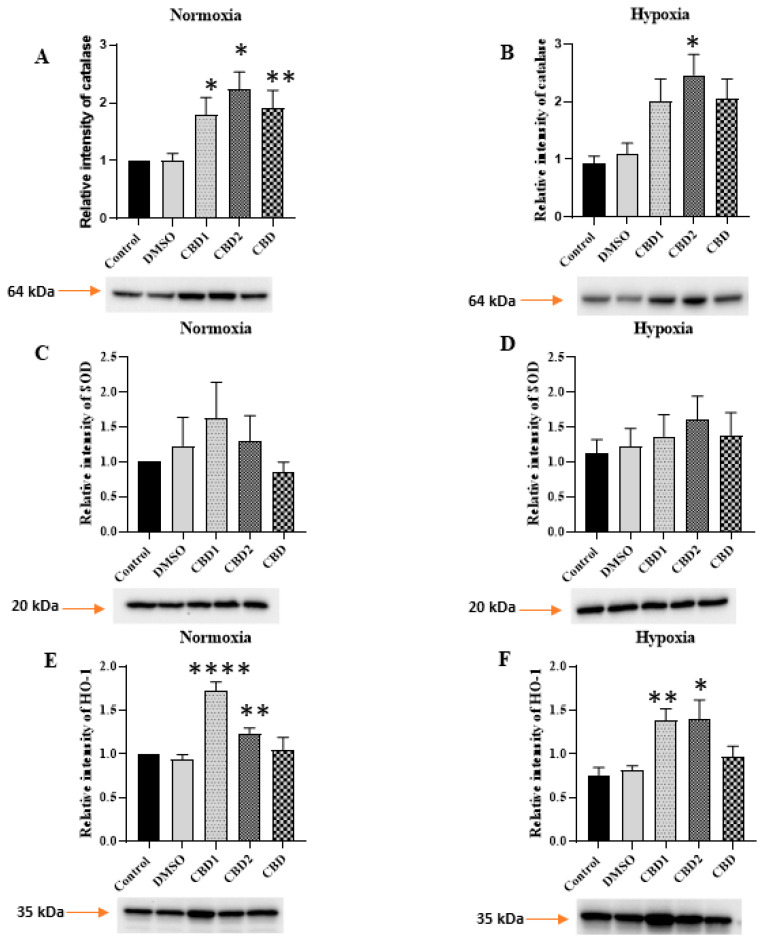
Effect of 10 μM of CBD1, CBD2, and CBD treatment on antioxidant enzymes. Alterations of protein level of catalase (**A**,**B**), SOD (**C**,**D**) and HO-1 (**E**,**F**) were evaluated by Western blot analysis under normoxic conditions (**A**,**C**,**E**) and hypoxia (**B**,**D**,**F**). Values were normalized to the total protein level and expressed as the mean ± SEM, n = 6–10. The significance of differences among groups was evaluated with *t*-test. *p* values of 0.05 or less were considered significant in each graph. *, **, and **** represent *p* < 0.05, *p* < 0.01, and *p* < 0.0001, respectively.

## Data Availability

The data that support the findings of this study are available from the corresponding author upon reasonable request.

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
