# Peer review of "Pharmacological Evaluation of Newly Synthesized Cannabidiol Derivates on H9c2 Cells"

_antioxidants, 2023, doi:10.3390/antiox12091714_

Round 1
Reviewer 1 Report
The manuscript does not look ready for publication and raises many questions.
Author postulated in Lines 90-92 that "Moreover, our plan was to investigate their cardioprotective effect and molecular mechanism during oxidative stress and simulated ischemia and reperfusion." However, the authors further write about hypoxia. Ischemia and hypoxia are not the same thing, and the authors need to clarify what they did.
Line 83 "its chemical transformation is only sporadically addressed" - However, the authors do not provide any references to these sporadic studies. the authors need to explain more clearly why these modifications were chosen. If there is no data on the effect of such modifications on the activity of CBD derivatives, it is necessary to use data on modifications of other compounds of other structural groups.
Section 2.2 The authors claim the synthesis of three CBD derivatives, but describe the synthesis and physico-chemical data of only one synthesized compound. Line 95: "The synthesis of ILKA652 and ILKA655 (Figure 1 A) will be published elsewhere". This is unacceptable. All data should be given for all newly synthesized compounds.
The treatment of H9c2 cells with H2O2 and pre-treatment with compounds did not adequately described as well as simulated hypoxia. All manipulation with H9c2 cells should be described in Methods section.
Figure 2. The data presented in Figure 2 are described as "H9c2 cells were treated with different concentrations of synthetic CBD derivatives and CBD 1 µM, 3 µM, 10 µM, 30 µM, 281 100 µM, 300 µM, 500 µM" but the graph has only six points of each compound' concentration presented as Log10. Thus, the graph does not match his description.
Section 3.3, Figure 3. The data presented in Figure 3 are no statistically significantly different and there is no reason to say that these substances have any significant positive effect on the viability of H2O2-treated H9c2 cells.
The cell viability in Figure 4a is presented as % but it is unclear what percentage was calculated, especially since the control cells have a viability of only 86%, which seems insufficient. Moreover, it is unclear why ILKA655 was not studied in tripan blue test.
Line 325 has a misleading: "duplicate (n = 3 experiments)" as well as Line 326 "n = 4-6 experiments". Exactly how many experiments and duplicates were carried out?
Section 3.5. There is no data on the use of any standard in the study of the antioxidant activity of substances. A known control should be used in such an experiment to evaluate the efficiency of the method at a minimum and the level of activity of substances as a maximum. Moreover, the substances were used only in one concentration.
I also have questions about the chosen method. To assess the direct antioxidant activity, there are several approaches, each of which has its own rationale and limitations. It is unclear why the authors chose only one method to analyze the effects of substances. For exampe, see Amorati, R.; Valgimigli, L. Advantages and limitations of common testing methods for antioxidants. Free radical research 2015, 49, 633-649, doi:10.3109/10715762.2014.996146.
Section 3.6 The absence of the substance ILKA655 in these studies is not clear.
In hypoxia, oxidative stress does develop, but the authors did not conduct any experiments to show that this happens in their case. Usually, this is done by testing the level of ROS or the level of lipid peroxidation in cells. Under oxidative stress, there is often (but not always) a decrease in the expression or activity of antioxidant enzymes. The authors do not analyze the differences in the level of expression in normoxia and hypoxia in order to somehow link these data.
The Discussion is very weak. The authors should discuss the structural features that ensure such activity of substances. It is necessary to analyze the influence of the structure on activity, including using literary data.
The authors need to analyze whether the cytoprotective effect of substances is a consequence of their direct antioxidant properties or their indirect antioxidant action. The authors show the effect of substances on antioxidant defense enzymes, but they do not discuss in any way how the substances under study can affect the level of these enzymes.
Author Response
First, we have to thank this reviewer for her/his constructive comments that she/he made about our manuscript. We tried to incorporate all of them. We believed that those significantly improved the quality of our revised manuscript.
1, The manuscript does not look ready for publication and raises many questions. Author postulated in Lines 90-92 that "Moreover, our plan was to investigate their cardioprotective effect and molecular mechanism during oxidative stress and simulated ischemia and reperfusion." However, the authors further write about hypoxia. Ischemia and hypoxia are not the same thing, and the authors need to clarify what they did.
We have modified the manuscript according to the suggestion of the reviewer.
2, Line 83 "its chemical transformation is only sporadically addressed" - However, the authors do not provide any references to these sporadic studies.
We inserted the following references into the manuscript:
Caprioglio, D., Mattoteia, D., Muñoz, E., Taglialatela-Scafati, O. & Appendino, G. One-Pot Oxidative Heterofunctionalization of Resorcinolic Cannabinoids to Non-thiophilic Aminocannabinoquinones. Eur. J. Org. Chem. e202101410, DOI: 10.1002/ejoc.202101410 (2022).
Caprioglio, D. Mattoteia, D. Pollastro, F., Negri, R., Lopatriello, A., Chianese, G., Minassi, A., Collado, J. A., Munoz, E., Taglialatela-Scafati, O. & Appendino G. The Oxidation of Phytocannabinoids to Cannabinoquinoids. J. Nat. Prod. 83, 1711–1715, https://doi.org/10.1021/acs.jnatprod.9b01284 (2020).
Ziegler, T. & Cosky E. Mitsonobu reaction of cannabidiol. Synthesis of water-soluble cannabidiol derivatives. Arkivoc, part iv, 198-205. DOI: https://doi.org/10.24820/ark.5550190.p011.347 (2021).
Jiang, X., Zhang, Z., Zuo, J., Wu, C., Zha, L., Xu, Y., Wang, S., Shi, J., Liu, X.-H., Zhang, J. & Tang, W. Novel cannabidiol−carbamate hybrids as selective BuChE inhibitors: Docking-based fragment reassembly for the development of potential therapeutic agents against Alzheimer's disease. Eur. J. Med. Chem. 223, 113735, DOI: https://doi.org/10.1016/j.ejmech.2021.113735 (2021).
The following text has been inserted after the mentioned sentence:
Most of the modifications published so far focus on improving the solubility, absorption and bioavailability of CBD, and little information is available on how chemical modifica-tions affect the biological activities of the synthetic derivatives compared to the parent compound. The study of derivatized cannabinoids is still in its infancy, and recent litera-ture reviews highlight that there is ample room for the development of new chemically modified cannabinoids and a great need for biological characterization of these new de-rivatives [23,24].
And the following references are added:
Breaking bud: the effect of direct chemical modifications of phytocannabinoids on their bioavailability, physiological effects, and therapeutic potential. DOI: 10.1039/d3ob00068k;
Millar, S. A.; Maguire, R. F.; Yates, A. S.; O’Sullivan, S. E. Towards better delivery of cannabidiol (CBD). Pharmaceuticals 2020, 13, 219(1)–219(15); Caprioglio, D.; Mattoteia, D.; Taglialatela-Scafati, O.; Muñoz, E.; Appendino, G. Cannabinoquinones: Synthesis and Biological Profile. Biomolecules 2021, 11, 991. https://doi.org/10.3390/biom11070991
The authors need to explain more clearly why these modifications were chosen.
The following text has been inserted into the Abstract:
„ However, CBD, like tetrahydrocannabinol (THC), has low bioavailability, poor water solubility, and a variable pharmacokinetic profile, which hinders its therapeutic use. Chemical derivatization of CBD offers us potential ways to overcome these issues. We prepared three new CBD derivatives substituted on the aromatic ring by Mannich-type reactions, which have not been described so far for the modification of cannabinoids, and studied the protective effect they have on cardiomyo-cytes exposed to oxidative stress and hypoxia/reoxygenation (H/R) compared to the parent compound.”
We reworded the last paragraph of the Introduction as follows:
The known chemical derivatizations of CBD can be classified into two main types: i) mod-ifications of the monoterpene ring [25] ii) modifications of the resorcinol moiety [19-22]. The latter modifications include oxidative conversion to cannabinoid quinones [19,20] and alkylation [21] of the hydroxyl groups or carbamate formation [21] with various rea-gents. The resorcinol unit of CBD is sensitive to oxidation, easily transforms into a quinoid structure, which, however, is not stable, can dimerize or undergo further oxidative degra-dation [19]. At the same time, the resorcinol structure offers the possibility of regioselective introduction of various substituents into the benzene ring, primarily substituents con-taining an amino group, by Mannich-type reactions [20,22]. Therefore, we decided to fol-low a new modification route that has not been applied to cannabinoids so far, the func-tionalization of the aromatic ring of CBD by Mannich reaction. With Mannich-type reac-tions, three different substitution patterns can be formed on phenols (Figure 1), the aro-matic ring can be functionalized with an alkylaminomethyl group (A), an oxazine (B) [20,22] or an alkoxymethyl group (C). Importantly, the presence of the amino groups in the compound-types A and B provides an opportunity for salt formation, which increases water solubility and thus bioavailability. In this work, we prepared 1-1 prototypes of all three substitution patterns achievable by Mannich reactions and investigated their cyto-protective effect and molecular mechanism during oxidative stress and hypoxia and re-oxygenation.
A new Figure is added:
Figure 1. Different functionalizations of phenols by Mannich-type reactions
References:
Hanus, L. O., Tchilibon, S., Ponde, D. E., Breuer, A., Fride, E., & Mechoulam, R. (2005). Enantiomeric cannabidiol derivatives: synthesis and binding to cannabinoid receptors. Organic & Biomolecular Chemistry, 3(6), 1116. doi:10.1039/b416943c
- Mechoulam, Z. Ben-Zvi, Y. Gaoni, Hashish—XIII: On the nature of the beam test, Tetrahedron, Volume 24, Issue 16, 1968, Pages 5615-5624, https://doi.org/10.1016/0040-4020(68)88159-1.
Omura, Y., Taruno, Y., Irisa, Y., Morimoto, M., Saimoto, H. & Shigemasa, Y. Regioselective Mannich reaction of phenolic compounds and its application to the synthesis of new chitosan derivatives. Tetrahedron Lett. 42, 7273-7275 DOI: https://doi.org/10.1016/S0040-4039(01)01491-5 (2001).
Burke, W. J., Murdock, K. C. & Ec, G. Condensation of Hydroxyaromatic Compounds with Formaldehyde and Primary Aromatic Amines. J. Am. Chem. Soc., 76, 1677–1679, DOI: https://doi.org/10.1021/ja01635a065 (1954).
3, If there is no data on the effect of such modifications on the activity of CBD derivatives, it is necessary to use data on modifications of other compounds of other structural groups.
Since there are no similar derivatives in the scientific literature, we cannot compare our synthesized derivatives with any other than the parent compound, CBD.
4, Section 2.2 The authors claim the synthesis of three CBD derivatives, but describe the synthesis and physico-chemical data of only one synthesized compound. Line 95: "The synthesis of ILKA652 and ILKA655 (Figure 1 A) will be published elsewhere". This is unacceptable. All data should be given for all newly synthesized compounds.
Synthesis of Ilka652 and Ilka655 are inserted.
5, The treatment of H9c2 cells with H2O2 and pre-treatment with compounds did not adequately described as well as simulated hypoxia. All manipulation with H9c2 cells should be described in Methods section.
The following sentences are inserted to the methods section:
For the measurement of antioxidant activity against H2O2-induced oxidative stress cells were seeded as described for the IC50 measurement. After 24h cells were treated with 0.4% DMSO, CBD1, CBD2, CBD3 or CBD. Then, cells also were treated with 125 μM H2O2 for 24h. Next day, cell viability was measured by MTT assay as described previously.
6, Figure 2. The data presented in Figure 2 are described as "H9c2 cells were treated with different concentrations of synthetic CBD derivatives and CBD 1 µM, 3 µM, 10 µM, 30 µM, 281 100 µM, 300 µM, 500 µM" but the graph has only six points of each compound' concentration presented as Log10. Thus, the graph does not match his description.
Thank you for your observation. We have modified. H9c2 cells were treated with different concentrations of synthetic CBD derivatives and CBD 1 µM, 3 µM, 10 µM, 100 µM, 300 µM, 500 µM
7, Section 3.3, Figure 3. The data presented in Figure 3 are no statistically significantly different and there is no reason to say that these substances have any significant positive effect on the viability of H2O2-treated H9c2 cells.
We rephrased the sentence to make it clear, that the difference was not statistically significant.
8, The cell viability in Figure 4a is presented as % but it is unclear what percentage was calculated, especially since the control cells have a viability of only 86%, which seems insufficient. Moreover, it is unclear why ILKA655 was not studied in tripan blue test.
It is written in Methods part 2.7. „The live cell count was divided by the total cell count and was multiplied by 100 the percentage viability. „
9, Line 325 has a misleading: "duplicate (n = 3 experiments)" as well as Line 326 "n = 4-6 experiments". Exactly how many experiments and duplicates were carried out?
These experimental numbers are related to different experiments.
Figure 4.A panel: trypan blue exclusion: The measurements were carried out in duplicate (n = 3 experiments).
Figure 4.B panel LDH activity: The measurements were carried out in triplicate. n = 4-6 experiments
10, Section 3.5. There is no data on the use of any standard in the study of the antioxidant activity of substances. A known control should be used in such an experiment to evaluate the efficiency of the method at a minimum and the level of activity of substances as a maximum. Moreover, the substances were used only in one concentration. I also have questions about the chosen method. To assess the direct antioxidant activity, there are several approaches, each of which has its own rationale and limitations. It is unclear why the authors chose only one method to analyze the effects of substances. For exampe, see Amorati, R.; Valgimigli, L. Advantages and limitations of common testing methods for antioxidants. Free radical research 2015, 49, 633-649, doi:10.3109/10715762.2014.996146.
We have inserted the following to the discussion:
Free radical scavenging ability of CBD is thought to contribute to its biological effect. In re-cent years, several studies have aimed to investigate the free radical scavenging capacity of CDB and other phytocannabinoids originated from Cannabis sativa [28-30]. However, recently Boulebd and colleagues showed that antioxidant activity of CBD is depends on the media of the experiments. They suggested that in aqueous phase at physiological pH CBD exhibits significant antioxidant activity, but in physiological lipid media the effect is almost diminished [31]. In our study we have used ABTS assay to compare antioxidant properties of CBD and its derivatives based on the following reasons. ABTS is commonly used method to measure antioxidant activity of CBD and phytocannabinoids and based on the previous results [28-30] and our own observation it is more sensitive for weaker an-tioxidants. In previous studies using different assays, the authors always found antioxi-dant activity for CBD. Although, there were differences between the antioxidant intensity based on the chosen assays but in all cases the antioxidant activity of CBD was compara-ble with vitamin E [32]. We have chosen ABTS assay and compared out new molecules to the parent molecule. Previously, it has been suggested that the antioxidant activity of CBD and phytocannabinoids rely on the phenolic groups, which can easily be converted to quinoid form [32,33]. Our results are in line with the literature, we found no antioxidant activity of CBD3, since the para substituents of phenolic groups prevent the transfor-mation of the molecule into the quinoid form. Also, CBD1 exhibit moderate antioxidant activity, which may relate to the [1,3]-oxazine group. However, the antioxidant effect of the new derivatives is lower than parent molecule CBD.
Please note, we compared the antioxidant activity of the new derivatives. The compounds were tested in 3 µM and 10 µM concentration. However, the lower concentration has moderate antioxidant activity, we depict only the 10 µM results.
11, Section 3.6 The absence of the substance ILKA655 in these studies is not clear.
Since during LDH assay and TAC measurements compound 655 (CBD3) showed negligible effect the further evaluation of this compound was stopped.
12, In hypoxia, oxidative stress does develop, but the authors did not conduct any experiments to show that this happens in their case. Usually, this is done by testing the level of ROS or the level of lipid peroxidation in cells. Under oxidative stress, there is often (but not always) a decrease in the expression or activity of antioxidant enzymes. The authors do not analyze the differences in the level of expression in normoxia and hypoxia in order to somehow link these data.
We have inserted the following sentence to the discussion.
There are some limitations of the current study. There were no direct ROS measurements during H/R experiments. However, it is not likely to happen that during 4 h hypoxia and 3 h reoxygenations is not sufficient injury to the cells, since it is a well-established model. Furthermore, additional studies with the current derivatives and new similar molecules need to be done to accurately demonstrate precise structure-activity relationships.
13, The Discussion is very weak. The authors should discuss the structural features that ensure such activity of substances. It is necessary to analyze the influence of the structure on activity, including using literary data. The authors need to analyze whether the cytoprotective effect of substances is a consequence of their direct antioxidant properties or their indirect antioxidant action. The authors show the effect of substances on antioxidant defense enzymes, but they do not discuss in any way how the substances under study can affect the level of these enzymes.
We rewrote the discussion part.

Reviewer 2 Report
The paper could be accepted after major revision.
Here are a several recommendations:
1. The tittle should be modified. Do not use abbreviations in the title. As well the parent compound is always under investigation. So, the title could be: Pharmacological evaluation of newly synthesized cannbidiol derivatives in rat cardiomyocytes H9c2.
2. Abstract should be completely rewritten.
I am not sure that the goal of the study could be formulated via using term “anti-ischemic effect”. Indeed, studying anti-ischemic effect suggest using in-vivo models. Authors evaluated effect of compound at hypoxic conditions in vitro only. Thus, the sentence on Lines 21-22 could be: Our goal was to increase the anti-hypoxic effect of the parent compound by synthetic modification of CBD.
The sentence (Lines 24-25) (MTT assay was performed to determine IC50 values and biocompatibility) should be specified. Indeed, term “biocompatibility” is usually applied to describe toxic/side effects of biomaterials, but not small molecules.
Thus, the sentence could be modified to: “MTT assay was performed to determine viability of rat cardiomyocytes treated by test compounds.”
I recommend changing the names of the compounds, like compounds 1-3 or CBD1-3. Current compound names look too complicated and remind lab codes.
What is TAC assay?
Lines 28-29: pIC50 values were: ILKA652: 4.113, ILKA653: 3.995, ILKA655: 4.190, and CBD: 4.671. What kind of activity? Please specify.
Introduction.
Authors must explain design of new compounds. Why were synthesized the compounds with these sub-molecular fragments?
Please, add references (lines 86-88) for the sentence “The resorcinol unit of CBD offers the possibility of regioselective introduction of various substituents into the benzene ring by Mannich-type reactions.”
Methods.
I think authors should describe a hypoxia/normoxia experiment in one section. Although, the cell viability tests (LDH and Trypan blue staining) could be described in another section.
Sections 2.9 and 2.10 should be joined in one section.
Results.
Fig. 1. There are not necessary to duplicate structures. So, Figure A1 should be deleted.
Figure 2. Legend. Please specify time of treatment (treated for 24 h…).
Section 3.3. Is the effect significant? It is not clear from Fig. 3.
Line 313. Authors used term “LDH liberation”. Does it mean that the compound 655 has cytotoxic effect? Why the compound 655 was not tested using trypan blue assay? If the compound 65 showed cytotoxicity by both methods, why MTT assay didn’t show cytotoxicity for this compound? Please, explain or retest.
Fig. 4B. Correct “Cytotoxicity” instead “Citotoxicity”.
Section 3.5. Why authors tested antiradical activity using ABTS, instead using more relevant ROS, like superoxide radical and hydrogen peroxide? Please explain in the paper.
Discussion:
Authors should add a hypothetical scheme for explanation/discussion effect of the test compounds on test cells.
Minor typos: see, for example Line 40.
Please, rephrase sentence on lines 44-46.
Use abbreviations (see line 59 etc).
Please, rephrase: “.. reductase has happened”.
Author Response
First, we have to thank this reviewer for her/his constructive comments that she/he made about our manuscript. We tried to incorporate all of them. We believed that those significantly improved the quality of our revised manuscript.
The paper could be accepted after major revision.
Here are a several recommendations:
- The tittle should be modified. Do not use abbreviations in the title. As well the parent compound is always under investigation. So, the title could be: Pharmacological evaluation of newly synthesized cannbidiol derivatives in rat cardiomyocytes H9c2.
Thank you for this suggestion. The title is modified as it was recommended.
- Abstract should be completely rewritten.
I am not sure that the goal of the study could be formulated via using term “anti-ischemic effect”. Indeed, studying anti-ischemic effect suggest using in-vivo models. Authors evaluated effect of compound at hypoxic conditions in vitro only. Thus, the sentence on Lines 21-22 could be: Our goal was to increase the anti-hypoxic effect of the parent compound by synthetic modification of CBD.
We have modified the abstract including the above-mentioned sentence.
The sentence (Lines 24-25) (MTT assay was performed to determine IC50 values and biocompatibility) should be specified. Indeed, term “biocompatibility” is usually applied to describe toxic/side effects of biomaterials, but not small molecules.
Thus, the sentence could be modified to: “MTT assay was performed to determine viability of rat cardiomyocytes treated by test compounds.”
We have accepted the suggestion.
I recommend changing the names of the compounds, like compounds 1-3 or CBD1-3. Current compound names look too complicated and remind lab codes.
It has been changed as suggested by the reviewer.
What is TAC assay?
Total antioxidant capacity assay. Now it is spell out.
Lines 28-29: pIC50 values were: ILKA652: 4.113, ILKA653: 3.995, ILKA655: 4.190, and CBD: 4.671. What kind of activity? Please specify.
pIC50 is the negative log of the IC50 value when converted to molar. Now it is spell out in lines 29
Introduction.
3, Authors must explain design of new compounds. Why were synthesized the compounds with these sub-molecular fragments?
The following text has been inserted after the mentioned sentence:
Most of the modifications published so far focus on improving the solubility, absorption and bioavailability of CBD, and little information is available on how chemical modifica-tions affect the biological activities of the synthetic derivatives compared to the parent compound. The study of derivatized cannabinoids is still in its infancy, and recent litera-ture reviews highlight that there is ample room for the development of new chemically modified cannabinoids and a great need for biological characterization of these new de-rivatives [23,24].
And the following references are added:
Breaking bud: the effect of direct chemical modifications of phytocannabinoids on their bioavailability, physiological effects, and therapeutic potential. DOI: 10.1039/d3ob00068k;
Millar, S. A.; Maguire, R. F.; Yates, A. S.; O’Sullivan, S. E. Towards better delivery of cannabidiol (CBD). Pharmaceuticals 2020, 13, 219(1)–219(15); Caprioglio, D.; Mattoteia, D.; Taglialatela-Scafati, O.; Muñoz, E.; Appendino, G. Cannabinoquinones: Synthesis and Biological Profile. Biomolecules 2021, 11, 991. https://doi.org/10.3390/biom11070991
The authors need to explain more clearly why these modifications were chosen.
The following text has been inserted into the Abstract:
„ However, CBD, like tetrahydrocannabinol (THC), has low bioavailability, poor water solubility, and a variable pharmacokinetic profile, which hinders its therapeutic use. Chemical derivatization of CBD offers us potential ways to overcome these issues. We prepared three new CBD derivatives substituted on the aromatic ring by Mannich-type reactions, which have not been described so far for the modification of cannabinoids, and studied the protective effect they have on cardiomyo-cytes exposed to oxidative stress and hypoxia/reoxygenation (H/R) compared to the parent compound.”
We reworded the last paragraph of the Introduction as follows:
The known chemical derivatizations of CBD can be classified into two main types: i) mod-ifications of the monoterpene ring [25] ii) modifications of the resorcinol moiety [19-22]. The latter modifications include oxidative conversion to cannabinoid quinones [19,20] and alkylation [21] of the hydroxyl groups or carbamate formation [21] with various rea-gents. The resorcinol unit of CBD is sensitive to oxidation, easily transforms into a quinoid structure, which, however, is not stable, can dimerize or undergo further oxidative degra-dation [19]. At the same time, the resorcinol structure offers the possibility of regioselective introduction of various substituents into the benzene ring, primarily substituents con-taining an amino group, by Mannich-type reactions [20,22]. Therefore, we decided to fol-low a new modification route that has not been applied to cannabinoids so far, the func-tionalization of the aromatic ring of CBD by Mannich reaction. With Mannich-type reac-tions, three different substitution patterns can be formed on phenols (Figure 1), the aro-matic ring can be functionalized with an alkylaminomethyl group (A), an oxazine (B) [20,22] or an alkoxymethyl group (C). Importantly, the presence of the amino groups in the compound-types A and B provides an opportunity for salt formation, which increases water solubility and thus bioavailability. In this work, we prepared 1-1 prototypes of all three substitution patterns achievable by Mannich reactions and investigated their cyto-protective effect and molecular mechanism during oxidative stress and hypoxia and re-oxygenation.
A new Figure is added:
Figure 01. Different functionalizations of phenols by Mannich-type reactions
References:
Hanus, L. O., Tchilibon, S., Ponde, D. E., Breuer, A., Fride, E., & Mechoulam, R. (2005). Enantiomeric cannabidiol derivatives: synthesis and binding to cannabinoid receptors. Organic & Biomolecular Chemistry, 3(6), 1116. doi:10.1039/b416943c
- Mechoulam, Z. Ben-Zvi, Y. Gaoni, Hashish—XIII: On the nature of the beam test, Tetrahedron, Volume 24, Issue 16, 1968, Pages 5615-5624, https://doi.org/10.1016/0040-4020(68)88159-1.
Omura, Y., Taruno, Y., Irisa, Y., Morimoto, M., Saimoto, H. & Shigemasa, Y. Regioselective Mannich reaction of phenolic compounds and its application to the synthesis of new chitosan derivatives. Tetrahedron Lett. 42, 7273-7275 DOI: https://doi.org/10.1016/S0040-4039(01)01491-5 (2001).
Burke, W. J., Murdock, K. C. & Ec, G. Condensation of Hydroxyaromatic Compounds with Formaldehyde and Primary Aromatic Amines. J. Am. Chem. Soc., 76, 1677–1679, DOI: https://doi.org/10.1021/ja01635a065 (1954).
Methods.
4, I think authors should describe a hypoxia/normoxia experiment in one section. Although, the cell viability tests (LDH and Trypan blue staining) could be described in another section. Sections 2.9 and 2.10 should be joined in one section.
Thank you for this suggestion. Now these are in one section 2.9.
Results.
5, Fig. 1. There are not necessary to duplicate structures. So, Figure A1 should be deleted.
It has been omitted.
6, Figure 2. Legend. Please specify time of treatment (treated for 24 h…).
Thank you for this suggestion Now time is written: „H9c2 cells were treated with different concentrations of synthetic CBD derivatives and CBD 1 µM, 3 µM, 10 µM, 100 µM, 300 µM, 500 µM for 24h”
7, Section 3.3. Is the effect significant? It is not clear from Fig. 3.
We rephrased the sentence to make it clear, that the difference was not statistically significant.
8, Line 313. Authors used term “LDH liberation”. Does it mean that the compound 655 has cytotoxic effect? Why the compound 655 was not tested using trypan blue assay? If the compound 65 showed cytotoxicity by both methods, why MTT assay didn’t show cytotoxicity for this compound? Please, explain or retest.
Thank you for your comments. We recheck of our all-raw datasets and also the analysis. We found that mistakenly one data belongs to hypoxic group instead of the normoxic results were used.
Since during LDH assay and TAC measurements compound 655 (CBD3) showed negligible effect the further evaluation of this compound was stopped.
9, Fig. 4B. Correct “Cytotoxicity” instead “Citotoxicity”.
It is corrected.
10, Section 3.5. Why authors tested antiradical activity using ABTS, instead using more relevant ROS, like superoxide radical and hydrogen peroxide? Please explain in the paper.
Free radical scavenging ability of CBD is thought to contribute to its biological effect. In re-cent years, several studies have aimed to investigate the free radical scavenging capacity of CDB and other phytocannabinoids originated from Cannabis sativa [28-30]. However, recently Boulebd and colleagues showed that antioxidant activity of CBD is depends on the media of the experiments. They suggested that in aqueous phase at physiological pH CBD exhibits significant antioxidant activity, but in physiological lipid media the effect is almost diminished [31]. In our study we have used ABTS assay to compare antioxidant properties of CBD and its derivatives based on the following reasons. ABTS is commonly used method to measure antioxidant activity of CBD and phytocannabinoids and based on the previous results [28-30] and our own observation it is more sensitive for weaker an-tioxidants. In previous studies using different assays, the authors always found antioxi-dant activity for CBD. Although, there were differences between the antioxidant intensity based on the chosen assays but in all cases the antioxidant activity of CBD was compara-ble with vitamin E [32]. We have chosen ABTS assay and compared out new molecules to the parent molecule. Previously, it has been suggested that the antioxidant activity of CBD and phytocannabinoids rely on the phenolic groups, which can easily be converted to quinoid form [32,33]. Our results are in line with the literature, we found no antioxidant activity of CBD3, since the para substituents of phenolic groups prevent the transfor-mation of the molecule into the quinoid form. Also, CBD1 exhibit moderate antioxidant activity, which may relate to the [1,3]-oxazine group. However, the antioxidant effect of the new derivatives is lower than parent molecule CBD.
Discussion:
11, Authors should add a hypothetical scheme for explanation/discussion effect of the test compounds on test cells.
We rewrite the discussion part.
12, Minor typos: see, for example Line 40.
13, Please, rephrase sentence on lines 44-46.
Thank you for this valuable suggestion, we reframed it as the follow:
In the last decades, there has been a growing interest in phytocannabinoid identified in Cannabis sativa.in components of Cannabis sativa, Cardinally CBD, nonpsychotropic cannabinoid, has attracted considerable attention for its multiple bioactivities beneficial to human health.
Use abbreviations (see line 59 etc).
14, Please, rephrase: “.. reductase has happened”.
It is rephrased.

Round 2
Reviewer 1 Report
The revised manuscript can be published. The final decision is at the discretion of the Editor.
Reviewer 2 Report
The paper could be accepted.
Please check typos before accepting (like "24h" should be separate).